# Clinical and Epidemiological Presentation of COVID-19 among Children in Conflict Setting

**DOI:** 10.3390/children9111712

**Published:** 2022-11-08

**Authors:** Maureen Dar Iang, Ola El Hajj Hassan, Maureen McGowan, Huda Basaleem, Khaled Al-Sakkaf, Albrecht Jahn, Fekri Dureab

**Affiliations:** 1Heidelberg Institute of Global Health, Hospital University, 69120 Heidelberg, Germany; 2Faculty of Medicine and Health Sciences, University of Aden, Madinat Al-Shaab, 63 O2, Aden P.O. Box 11011, Yemen

**Keywords:** COVID-19, SARS-CoV-2, children, equity, Yemen

## Abstract

Background: This study aims to describe the observable symptoms of children with COVID-19 infection and analyze access to real-time polymerase chain reaction (RT-PCR) testing among children seeking care in Yemen. Method: In the period of March 2020–February 2022, data were obtained from 495 children suspected to have been infected with COVID-19 (from a larger register of 5634 patients) from the Diseases Surveillance and Infection Control Department at the Ministry of Public Health and Population in Aden, Yemen. Results: Overall, 21.4% of the children with confirmed COVID-19 infection were asymptomatic. Fever (71.4%) and cough (67.1%) were the most frequently reported symptoms among children, and children were less likely to have fever (*p* < 0.001), sore throat (*p* < 0.001) and cough (*p* < 0.001) compared to adults. A lower frequency of COVID-19-associated symptoms was reported among children with positive RT-PCR tests compared to children with negative tests. A lower rate of testing was conducted among children (25%) compared to adults (61%). Fewer tests were carried out among children <5 years (11%) compared to other age groups (*p* < 0.001), for children from other nationalities (4%) compared to Yemeni children (*p* < 0.001) and for girls (21%) compared to boys (30%) (*p* < 0.031). Conclusion: Understanding and addressing the cause of these disparities and improving guidelines for COVID-19 screening among children will improve access to care and control of the COVID-19 pandemic.

## 1. Introduction

With more than 617.59 million confirmed cases and 6.53 million deaths attributed to the COVID-19 pandemic, caused by the severe acute respiratory syndrome coronavirus 2 virus (SARS-CoV-2), COVID-19 has had a global impact [1]. The World Health Organization (WHO) declared COVID-19 a pandemic in March 2020 and provided guidance for case management and control of SARS-CoV-2 infection. The initial phase of the pandemic reported SARS-CoV-2 infections to be concentrated among middle-aged and older adults, while the latter part of the pandemic reported infections among all age groups—including mild infections among children [2,3]. Though COVID-19 affects all age groups and communities, a higher exposure to and a higher incidence of COVID-19 are reported among resource-limited settings and vulnerable populations [4,5,6]. Disparities in health service access—in part attributed to socio-economic factors—and resulting poor health outcomes are well documented globally and in Yemen [7,8]. These disparities were further exacerbated during the COVID-19 pandemic due to lockdown and COVID-19 control measures, described as a “syndemic pandemic” [9].

A large majority of people with COVID-19 were observed to have mild or asymptomatic infections, with primary symptoms of COVID-19 including fever, cough, sore throat, shortness of breath, fatigue, loss and/or change of taste/smell and headaches developing 2–14 days post-exposure to the virus. The WHO recommends confirming the diagnosis of COVID-19 using real-time polymerase chain reaction [RT-PCR] in suspected cases, based on clinical symptoms and/or a history of contact with a confirmed case. COVID-19 affects most systems in the body, as supported by a recent systematic review and meta-analysis which reported on more than 96 symptoms attributed to COVID-19 infection among all age groups [10]. However, the symptoms observed in children varied widely, and were often non-specific [10,11,12]. The experienced symptoms and severity of COVID-19 outcomes vary based on the prevailing variant of SARS-CoV-2 and by patient vaccination status [13]. Serious illness and death were typically higher among adults and children suffering from comorbid diseases and conditions (e.g., hypertension, diabetes, chronic lung diseases, obesity, and heart disease) [14,15]. While most deaths were a direct result of COVID-19 infection, excess deaths during the pandemic were attributed to limited access to healthcare services and timely intervention—a challenge both globally and locally in the context of Yemen [16,17].

Yemen, with 21 governorates and one municipality, has a population of ~30 million people, approximately 70% of whom live in rural areas. Following the civil war in 2014, health facilities were damaged and/or closed and only 50% are partially functional with a reduced capacity for providing healthcare services [18]. A major ongoing concern in the country is security, which also impedes access to healthcare services. A city in south-east Yemen which maintains better accessibility and availability of healthcare services is Aden [19]. The Aden Governorate of Yemen declared a health emergency on 11 May 2020 and reported 354 confirmed COVID-19 cases and 84 deaths by 1 June 2020. Yemen has since experienced four waves of COVID-19 with 11,935 confirmed cases and 2158 reported deaths by October 2022 [1]. With limited functioning health facilities, Yemen faces challenges providing the health services necessary to tackle COVID-19 infection, particularly large-scale RT-PCR testing [20]. An online survey conducted among health workers in Yemen on health system capacity and readiness to manage the COVID-19 pandemic found that the majority of health workers believed the system was not adequately prepared or that it did not have the resources and capacity to manage the COVID-19 pandemic. More specifically, more than 80% of health workers rated the availability of isolation rooms, diagnostic devices, intensive care rooms, as well as ventilators in their health care facilities as very poor or poor [21]. While little research has been conducted in Yemen to understand barriers to healthcare access in the context of COVID-19, access to women’s health services and Malaria treatment indicated perceived quality of care, restriction on women’s mobility, distance to health services and cost of services as additional barriers to healthcare access [8,22].

This study aims to analyze the presenting symptoms of children with COVID-19 infection and to understand access to confirmatory RT-PCR testing among the children. Our findings aim to inform the national guidelines for the case identification and management of RT-PCR testing to ensure better access to COVID-19 healthcare services for children in Yemen.

## 2. Materials and Methods

This retrospective study was part of a larger study investigating the COVID-19 pandemic in Aden, Yemen, with the objective to provide evidence for an improved and locally adapted COVID-19 control strategy. This study included 495 children of ≤18 years from a larger dataset of 5634 cases. Study participants’ clinical and epidemiological data were obtained in the period of March 2020–February 2022 from the Department of Diseases Surveillance and Infection Control, at the Ministry of Public Health and Population in Aden. The RT-PCR tests were conducted at the Central Laboratory of Aden. Data were collected by healthcare workers in hospitals and entered into the surveillance system. Data were collected in accordance with WHO guidelines including socio-demographic data (e.g., gender, age, address, nationality, occupation); date of symptom onset; history of contact with COVID-19 confirmed cases within the last 14 days; medical history of chronic illnesses (e.g., heart disease, high blood pressure, diabetes mellitus, chronic lung disease, HIV/AIDS, and kidney disease); hospital admission; RT-PCR test result (when available); presenting symptoms at the time of screening (reported by healthcare workers); and patient outcomes.

All the children (ages ≤ 18 years) suspected of COVID-19 infection, among 5634 recorded cases, were included in the study. Inclusion criteria: children (ages ≤ 18 years) registered in the surveillance system for seeking COVID-19 health services. Among 495 children suspected of COVID-19 infections, symptoms were recorded for all children and RT-PCR test results were available for 25% of children (126/495). Among 70 children with confirmed COVID-19 infection (70/126; 55%), hospital admission status was available for all children (N = 70) and final outcome status was available for 69 children (99%).

An RT-PCR test was conducted in alignment with WHO guidelines, which have been adopted by Yemen. During early 2020, the WHO developed case definitions of COVID-19 (suspected/possible, probable and confirmed cases) based on a range of clinical and epidemiological criteria (as listed below) (version from 16 December 2020) [23]. This case definition was used by healthcare workers screening patients for RT-PCR need.

Suspected COVID-19 cases: A person who meets the clinical and epidemiological criteria

Clinical criteria: Acute onset of fever and cough; or acute onset of any of three or more of the following symptoms including fever, cough, general weakness/fatigue, headache, myalgia, sore throat, coryza, dyspnea, anorexia/nausea/vomiting, diarrhea, altered mental status;Epidemiological criteria: residing or working in an area with high risk of transmission of virus: closed residential settings, humanitarian settings such as camp and camp-like settings for displaced persons; anytime within the 14 days prior to symptom onset; or residing or travelling to an area with community transmission anytime within the 14 days prior to symptom onset; or working in any health care setting, including within health facilities or within the community; anytime within the 14 days prior to symptom onset.A patient with severe acute respiratory illness (SARI): acute respiratory infection with history of fever or measured fever of ≥38 °C and cough; with onset within the last 10 days; and requires hospitalization).

Probable COVID-19 cases:

(1) A patient who meets clinical criteria above and is a contact of a probable or confirmed case, or epidemiologically linked to a cluster with at least one confirmed case. (2) A suspect case with chest imaging showing findings suggestive of COVID-19 disease based on chest radiography, chest CT scan and lung ultrasound. (3) A person with recent onset of anosmia (loss of smell) or ageusia (loss of taste) in the absence of any other identified cause. (4) Death, not otherwise explained, in an adult with respiratory distress preceding death AND was a contact of a probable or confirmed case or epidemiologically linked to a cluster with at least one confirmed case.

Confirmed COVID-19 cases:

A person with laboratory confirmation of COVID-19 infection, irrespective of clinical signs and symptoms.

After data collection, the investigators translated the data into English (from Arabic), which were then entered into an Excel sheet. All of the data were coded, labelled, and grouped into categories using Stata 13.1 software. Stata 13.1 was also used for data analysis including univariate analyses (e.g., frequencies, means, percentage, and standard deviations) to measure descriptive statistics and inductive analyses. Significance between variables was confirmed through the analysis of variance and chi-squared tests using the significance level of *p* < 0.05. Only salient results are presented in the results section below.

## 3. Results

### 3.1. Participants’ Characteristics

In this study, 495 COVID-19 suspected children patients ages 18 years and below were included. Table 1 provides an overview of participants’ socio-demographic and health characteristics. Over half of the participants (257/495; 51.9%) were male and 48.1% (238/495) were female. More than half of the children suspected of COVID-19 infection (266/495; 54%) were aged 10.1–18 years, 21% were aged 5–10 years, and a quarter (124/495; 25%) were below 5 years of age. The majority of participants (481/495; 97%) lived in Aden governorate, many of whom lived in Dar Sad (346/495; 69.9%) and in Seera (67/495; 13.5%). The majority of the suspected children were previously healthy, while few (18/495; 2.9%) reported non-communicable chronic diseases. Among children with confirmed COVID-19 infection (N = 70), 61% were male and 39% were female. The mean age of children with confirmed COVID-19 was 12.57 ± 4.9 years. No RT-PCR tests were conducted among children under one year of age. Children with confirmed COVID-19 infection had co-morbid chronic diseases in 4.2% of cases.

### 3.2. Clinical Presentation among Children

Among 495 children, 25% (N = 126) were tested using the RT-PCR test for the confirmation of SARS-CoV-2 infection. Meanwhile, 61% (N = 2967) adults with suspected COVID-19 (from the larger data register) were tested using RT-PCR tests. A total of 70 COVID-19 infections were confirmed among children (≤18 years) and 2157 among adults (>18 years) for a total 2227 patients, as shown in Table 2. In the entire data registry, children accounted for 3.1% of all COVID-19 confirmed cases (70/2227). Of children with confirmed COVID-19 infection, 21.4% (15/70) were asymptomatic. Only two of the children reported recent contact with a confirmed case of COVID-19.

Of those with symptoms (55/70; 79%), the number of symptoms reported was in the range of 1–8 symptoms (average 3.7 symptoms/child). The most common symptoms reported among children were fever (71.4%), cough (67.1%) and difficulty in breathing (44.3%). Other symptoms reported among children were chest pain (42.9%), muscles and/or joint pain (42.9%), sore throat (40%), and rhinorrhea (40%). In adults, the most common symptoms reported were fever (86.1%), cough (82.2%), sore throat (72.4%), and difficulty in breathing (53.5%). Adults were more likely to present with fever (*p* < 0.001), sore throat (*p* < 0.001) and cough (*p* = 0.001) compared to children. Patients also reported headache, loss of smell, or alternated taste among both children (7.1%, 2.9% and 2.9%) and adults (8.7%, 1.9% and 0.7%), Table 3.

COVID-19-associated symptoms (e.g., fever, cough, sore throat, chest pain, headache) were more frequently reported among children with negative PCR tests (N = 56) compared to children with positive PCR tests (N = 70). However, some symptoms including a nasal discharge, difficulty in breathing, muscles and/or joint pain, loss of smell and altered sense of smell were more commonly reported among children with positive RT-PCR tests, as shown in Table 4. Of children with confirmed COVID-19 cases, one child was admitted to the hospital and two deaths were recorded.

### 3.3. Identification and Diagnosis of Children with Suspected COVID-19 Infection

Of the children suspected to have COVID-19 infection, 25% (126/495) were tested using RT-PCR, whereas 61% of adults were tested. The proportion of suspected children tested for COVID-19 increased from 17% in 2020, to 32% in 2021, and 63% in 2022. Around 55% (70/126) of all tested children were confirmed positive for COVID-19, 3.1% among total reported positive cases (70/2227) (see Table 2). Those who received RT-PCR testing were compared against demographic characteristics. Children <5 years were significantly less likely to be tested for COVID-19 compared to other age groups (≤5 years = 11.3%; 6–10 years = 23.1%; ≥10.1 years = 33.4%; *p* < 0.001). Yemeni children were significantly more likely to be tested for COVID-19 compared to children of other nationalities (Yemeni = 35.3%, other nationality = 4.0%; *p* < 0.001). Finally, male children were significantly more likely to be tested compared to female children (29.7% male, 21.2% female; *p* = 0.031) (see Table 5).

## 4. Discussion

This paper presents descriptive data on the clinical and epidemiological characteristics of COVID-19 infection among children in the Aden governorate, during May 2020–February 2022.

Our findings show that the proportion of children who were confirmed COVID-19 positive (using RT-PCR testing) was 3.1% of total confirmed cases, thereby being much lower than reported proportion in other countries [3,24,25]. During the early pandemic period, the reported proportion of children with COVID-19 infection was lower compared to other age groups but the proportion varied from country to country. During the early reporting period, the proportion of children among total COVID-19 confirmed cases was documented at 2% in China (until 11 February 2020) [24], 1.7% in the USA (during 12 February 2020–2 April 2020) [25] and 5.2% in the USA (22 January–30 May 2020) [3]. Later in the pandemic period, the proportion of children (0–19 years) among the total COVID-19 confirmed cases was determined to be 20.5% globally [26], 14.7% among children under 15 years in Europe [27] and 18.4% among children under 19 years in the United States of America [28]. The prevalence of COVID-19 among children (0–17 years) in Nepal was 6.3 per 100,000 population, compared to 29.9 per 100,000 in the total population in the country [29].

The lower proportion of children confirmed with COVID-19 in our study population (3.1%) may be attributed to various reasons including healthcare-seeking behavior. Our study found that fewer children sought COVID-19 care compared to adult populations as indicated by 8.8% (495/5634) of the total data registry being children < 19 years old. As noted in previous literature on access to women’s health services and access to Malaria and Tuberculosis treatment in Yemen, perceived quality of care, women’s mobility, distance, and cost related to health care access were found to be important drivers of healthcare seeking and access in Yemen [8,22,30]. Similarly, our study found low rates of RT-PCR testing among children (25%) and high positivity rates (55%) which may indicate the limited availability of RT-PCR testing in Yemen [20,21], prioritization of severe adult cases as per national guidelines, or limited awareness of COVID-19 testing at the community level. Moreover, the lower sensitivity of RT-PCR testing among children may be another contributing factor. For example, a study in Norway found lower sensitivity of RT-PCR testing among children [31] and a systematic review (including 32 studies) reported that false-negative RT-PCR test results varied in the range of 2–58%, with an overall false negative rate of 10% among children and 12% among adults [32]. The false-negative rates varied based on the time between the onset of symptoms and the date the test was performed. However, a study in Washington (USA) found no significant difference in Cycle threshold (Ct) values of the RT-PCR test between symptomatic children and adults [33]. Additionally, challenges with sample collection, especially among young children, cannot be ruled out. Given the lack of widely accessible testing, the gaps in care for children who did not have access to RT-PCR testing should not be under-estimated. Consequently, the role these children may have played in the onward transmission of SARS-CoV-2 is of concern to the wider community.

The percentage of asymptomatic children in our study is 21.4%, which mirrors the rates found in recent systematic reviews [11,12,34,35] and case studies in China [36] and Mexico [37]. The reported percentage of children with asymptomatic COVID-19 infections in these studies was 13.1–28.4%. The most frequent symptoms reported among children were fever, cough, and difficulty in breathing and were often reported to a lower degree compared to the adult population. In our study, fever and cough among children were 71.4% and 67.1% at the time of presentation, respectively. These were followed by difficulty in breathing (44.3%), muscles and/or joint pain (42.9%), chest pain (42.9%), sore-throat (40%) and nasal discharge (40%). Our findings reported COVID-19 symptoms similar to those reported in case studies in Mexico [37], China [36] and the USA [38] as well as in systematic reviews [11,34,35]. In these case series, the level of reported fever among symptomatic children with COVID-19 was in the range of 69–78% and a cough was reported in 36–67% of cases. Compared to adult populations, children in our study reported lower frequency for most symptoms, which was also in line with other studies comparing frequency of symptoms among children and adults [39,40].

The WHO’s and Yemen’s guidelines for screening children for COVID-19 include presenting symptoms and history of recent travel to epidemic areas and/or contact with a confirmed case. The predictability of individual symptoms for the diagnosis of COVID-19 (positive RT-PCR test) and for the radiological finding of pneumonia has been reported in studies among both children and adults [35,41]. Four symptoms that were found to be strongly associated with a positive RT-PCR likelihood ratio (LR) in children were anosmia/ageusia at 7.33, nausea/vomiting at 5.51, headache at 2.49 and fever at 1.68 [41]. Fever at a sensitivity of 60.3%, cough at 47.4%, rhinorrhea at 21.1% and abdominal symptoms at 10.3% were found to be predictors for the radiological finding of pneumonia [35]. Contrarily, our results showed a higher frequency of COVID-19-associated symptoms among RT-PCR negative children compared to positive children.

A higher frequency of reported symptoms among the 56 children with negative RT-PCR tests could be attributed in part to false-negative RT-PCR tests, as previously discussed [31,32]. Moreover, patients may have other respiratory virus infections which should be considered and tested for. Symptoms of COVID-19, especially among children, are often non-specific [11,12], making it difficult to differentiate symptoms between COVID-19 and other respiratory viruses prevailing in the area. Various viruses including influenza virus, rhinovirus, respiratory syncytial virus and respiratory adenovirus, among others, are known to cause upper respiratory tract infection, with symptoms such as coryza, cough, sore throat, hoarseness, and fever with some progressing to lower respiratory tract infections [42]. The recent literature reported that in any flu season up to 9.8% of children aged 0–14 years old can have influenza, with a higher incidence reported among children under five years [43]. In fact, incidence of influenza among children under five years is estimated to be 110 million episodes per year [44]. A higher frequency of symptoms (e.g., fever, cough, runny nose, headache, myalgia, and sore throat) have previously been reported among children with influenza compared to COVID-19 [45,46]. However, a systematic review of 12 studies found that while no symptoms are characteristic of COVID-19 or influenza, diarrhea was more frequent among children with COVID-19 compared to influenza, which may be a discriminating factor between COVID-19 and other respiratory viruses [47]. Neurological symptoms such as headache and change in smell and taste are more frequent among adults with COVID-19 than with influenza; subsequently, the presence of these symptoms can be a differentiating factor for COVID-19 from influenza [48]. Reviews of similar studies reported above among children did not report on changes in smell and taste among children [47]. Our study found increased reports of loss of smell and altered sense of taste among children with an RT-PCR positive test; however, the numbers are too small to show a significant difference. The COVID-19-associated symptoms identified in our study (or lack thereof) and evidence of difficulty in differentiating COVID-19 and other respiratory viral infections [47] raises the question of whether the existent national guidelines using symptom-based criteria to determine the need for RT-PCR testing are appropriate for managing COVID-19 testing among children in Yemen.

Children are more likely to be asymptomatic or to have a mild COVID-19 infection compared to adults—as is supported by our study findings [39,40]; however, children face a similar risk of being infected with COVID-19 [49]. Our study found that young children, female children, and non-national children were less likely to be tested for COVID-19 using RT-PCR testing. This is particularly critical to consider as other recent studies have found that minority populations (including persons in low resourced areas) are at increased risk of COVID-19 infection [50] and of severe COVID-19 illness [51]. For example, children under the age of 4 years are at increased risk of severe COVID-19 pneumonia infection [35]. Moreover, a recent systematic analysis reported that mortality risk by age group is a J-shaped curve, with COVID-19 mortality being highest in both very young and older age groups; the lowest risk of mortality is recorded at age 7. Children under five years old are at higher risk of mortality compared to children 10–14 and 15–19 years of age [52]. Mortality associated with younger age groups has been particularly observed in South Asia, Sub-Saharan Africa and the Middle East [52]. Finally, the literature also indicates that while male children are at higher risk of COVID-19 infection and mortality, only a slightly higher incidence among male children is reported [26], while our findings show 61% males and 39% females among COVID-19 confirmed children. These findings raise concerns about the equitable access to COVID-19 testing services while children have equal risk of COVID-19 infection compared to the adult population [48] and higher incidence of COVID-19 infection reported among certain groups of population [4,5,6]

It is clear that the current case definition and testing guidelines are likely to miss cases of COVID-19 in children; a specific strategy for COVID-19 testing should therefore be adapted. This is demonstrated in numerous case series and systematic analyses which indicate an absence of symptoms specific to COVID-19 [11,12], the difficulty to differentiate COVID-19 from other respiratory viral infections including influenza [45,46,47], the increase in the risk of infection and death associated with COVID-19 among certain populations and younger children [4,5,6,52], as well as our observed findings of a higher frequency of symptoms among children with negative RT-PCR tests. There is also evidence of lower RT-PCR testing among children and particular groups of children (young age, female, children of other nationalities), highlighting inequity in access to COVID-19 testing.

This study is not without limitations. First, patients’ clinical data were collected from a surveillance system, and thus, the description of the symptoms is subjective, and no complete clinical picture is available. For example, it is unclear which criteria were used to test asymptomatic children for COVID-19. Second, the ability for the health system to provide diagnostic testing in the early stages of the pandemic (particularly in 2020) was very limited, and thus most suspected cases were not tested—this may have skewed the number of non-confirmed cases. Third, missing information regarding the final outcomes of children may create bias in the interpretation of our findings. Finally, we are not able to verify reasons for not offering an RT-PCR test to vulnerable groups (e.g., infants, under the age of five, female children, non-national children)—which calls for future research.

## 5. Conclusions

In conclusion, our study adds to the literature on COVID-19 presentation and testing among children in areas with limited access to healthcare services. Low rates of symptomatic children with confirmed COVID-19, and limited access to COVID-19 testing among specific groups of children raises the question to the appropriateness of currently used symptom-based identification criteria for children suspected of having COVID-19. This study also highlights inequities in access to COVID-19 testing services. Limited RT-PCR testing among children in general, female children and non-national children raises the issue of access to health services for vulnerable communities. Understanding and addressing the reasons for low rates of RT-PCR testing among children and specific groups of children will improve access to care and better control the COVID-19 pandemic. Ensuring access to necessary COVID-19 testing is vital not only for early detection and linkage to care, but for the prevention of future COVID-19 transmission.

## Figures and Tables

**Table 1 children-09-01712-t001:** Sociodemographic and Health Characteristics of COVID-19 Suspected Children.

	N = 495	%
Gender		
Female	238	48.1
Male	257	51.9
Age		
<5	124	25.1
5–10	105	21.2
10.1–18	266	53.7
Governorate Residence		
Aden	481	97.2
Other	14	2.8
District		
Dar Sad	346	69.9
Seera	67	13.5
Al Mualla	15	3.0
Ash Shaikh Outhman	13	2.6
Al Mansura	12	2.4
Khur Maksar	12	2.4
Al Buraiqeh	12	2.4
Attawahi	4	0.8
Presence of chronic diseases		
Diabetes Mellitus	5	1.0
High Blood Pressure	5	1.0
Cardiac diseases	2	0.4
Chronic Respiratory diseases	2	0.4
Blood Diseases	2	0.4
Renal Disease	1	0.2
Liver Disease	1	0.2

**Table 2 children-09-01712-t002:** Number of Suspected, Tested and COVID-19 Positive Cases.

	Suspected Cases	Tested Cases	Positive PCR-Test	Asymptomatic Positive
Adults	5139	2967	2157	150
Children	495	126	70	15
Total	5634	3093	2227	165

**Table 3 children-09-01712-t003:** Comparison of Reported Symptoms in RT-PCR Confirmed Cases by Age Group.

Symptoms					
	Children N = 70	%	Adults N = 2157	%	*p*-Value
Fever					<0.001
No	20	28.6	299	13.9	
Yes	50	71.4	1858	86.1	
Sore throat					<0.001
No	42	60.0	595	27.6	
Yes	28	40.0	1562	72.4	
Cough					0.001
No	23	32.9	383	17.8	
Yes	47	67.1	1774	82.2	
Nasal discharge					0.062
No	42	60.0	1518	70.4	
Yes	28	40.0	639	29.6	
Difficulty in breathing					0.126
No	39	55.7	1002	46.5	
Yes	31	44.3	1155	53.5	
Headache					0.654
No	65	92.9	1970	91.3	
Yes	5	7.1	187	8.7	
Chest pain					0.099
No	40	57.1	1017	47.1	
Yes	30	42.9	1140	52.9	
Muscles and/or Joint pain					0.26
No	40	57.1	1502	69.6	
Yes	30	42.9	655	30.4	
Diarrhea					0.794
No	69	98.6	2117	98.1	
Yes	1	1.4	40	1.9	
Loss of small					0.590
No	68	97.1	2115	98.1	
Yes	2	2.9	42	1.9	
Altered sense of taste					0.052
No	68	97.1	2141	99.3	
Yes	2	2.9	16	0.7	

**Table 4 children-09-01712-t004:** Symptoms of Children, Age 18 or below, with Suspected and Confirmed COVID-19 Infection.

	Reported Symptoms, Children with Suspected COVID-19 (N = 56)		Reported Symptoms, Children with Confirmed COVID-19 (N = 70)	
	N	%	N	%
Fever	44	78.6	50	71.4
Sore throat	36	64.3	28	40.0
Cough	45	80.4	47	67.1
Nasal discharge	17	30.4	28	40.0
Difficulty in breathing	22	39.3	31	44.4
Headache	11	19.6	5	7.1
Chest pain	28	50.0	30	42.9
Muscle and/or Joint pain	18	32.1	30	42.9
Diarrhea	2	3.6	1	1.4
Loss of smell	1	1.8	3	2.9
Altered sense of taste	1	1.8	2	2.9

**Table 5 children-09-01712-t005:** RT-PCR Testing among Children of Different Demographic Groups.

Demographics					
	RT-PCR Test Not Done (N)	%	RT-PCR Test Done (N)	%	*p*-Value
Age					<0.001
≤5	110	88.7	14	11.3	
6–10	80	76.9	24	23.1	
>10.1	175	66.5	88	33.4	
Nationality					<0.001
Yemeni	220	64.7	120	35.3	
Others	146	96.0	6	4.0	
Sex					0.031
Male	180	70.3	76	29.7	
Female	186	78.8	50	21.2	

## Data Availability

The data presented in this study are available on request from the corresponding author. The data are not publicly available due to privacy.

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
