# Peer review of "Clinical and Epidemiological Presentation of COVID-19 among Children in Conflict Setting"

_children, 2022, doi:10.3390/children9111712_

Round 1
Reviewer 1 Report
Thank you for the opportunity to review this very interesting article.
I congratulate the authors for this very valuable research, I am sure that the results of this study will help the scientific community in understanding COVID-19 in children and an important feature of this study is that it also reveals the problems associated with social factors which may cause inequality regarding access to health services.
In this study conducted from Yemen, the authors presented clinical, laboratory, and epidemiological data on pediatric cases of COVID-19. The most common symptoms were fever and cough. The findings were consistent with previous studies, which demonstrated that children of all ages are susceptible to SARS-CoV-2 infection and the majority of cases are asymptomatic or mild.
This manuscript has some issues that need to be addressed before it is considered for publication.My recommendations are stated in the text and highlighted. Please check the numbers and equations in the tables. I hope my comments help the authors to improve the quality of this manuscript.

Author Response
Thank you for your kind feedback and comments given to our manuscript with suggestions to improve the manuscript. We have revised based on your comments. Kindly find our answers below and revisions done based on your comments.
- Methodology section – What were the testing criteria for COVID?
We thank the reviewer for this comment. We have added our testing criteria for COVID-19; which reflects that of the 2022 “WHO COVID-19 Case Definition” (https://www.who.int/publications/i/item/WHO-2019-nCoV-Surveillance_Case_Definition-2020.1)
The methodology now includes definitions for “suspected COVID-19 cases”, “probable COVID-19 cases”, and “confirmed COVID-19 cases”. These changes can be viewed in lines 112-143.
- PCR test result (where appropriate) - I did not understand what you meant by where appropriate.
This change has been made. We have changed the sentence to reflect that the inclusion of the PCR test result in analysis when it was available in the surveillance system data. The sentence now reads as follows:
(Methodology, line 102-103):
“…hospital admission; RT-PCR test result (when available)…”
- The numeric data, normally distributed must be indicated with mean±SD, not range
We have now revised numeric data to include the mean and standard deviation rather than the range. This is reflected in the results section when we report on the demographic “age”.
(Results, lines 167-168)
“The mean age of children with confirmed COVID-19 was 12.57±4.9 years.”
- None of the children under one were tested with PCR test. Why were the children under 1 year not tested?
Unfortunately, data was not collected regarding reasons for not testing children under the age of one. We do however, value this comment and would be interested to pursue this in future research. We have now included this as a limitation section to our study in the discussion section.
(Discussion, lines 350-353)
Finally, we are not able to verify reasons for not offering RT-PCR test to vulnerable groups (e.g., infants, under the age of 5, female children, non-national children)-which calls for future research.
- 21.4% children with PCR positive tests were asymptomatic. Why were these cases tested? Did they have history of contact?
We found that only 2/70 children with confirmed COVID-19 reported to have a history of contact with confirmed COVID-19. Both cases are found to be asymptomatic. Unfortunately, the reasons why other 13 asymptomatic children were tested with RT-PCR is not available.
In the results section, we have now included the information for the two children, for whom reports do exist:
(Results, lines 179-180)
Only two reports identified why asymptomatic children had been tested for COVID-19, both of whom had recent contact with a confirmed case of COVID-19.
Additionally, we have added the above point as an additional limitation to the study in the discussion section:
(Discussion, lines 345-346)
“For example, it is unclear which criteria was used to test asymptomatic children for COVID-19 (with the exception of the two aforementioned highlighted cases).”
- There were 1-8 symptoms in patients. But here is written more than 25 symptoms. Do you mean something else?
Most children experienced between 1 and 8 symptoms. However, across all children, 25 different symptoms were recorded. To avoid confusion, we have now deleted the sentence from the manuscript.
- These symptoms are not specific for COVID 19 and may be seen in many different viral infections. These patients were both may be false negative or having other viral infections. I also recommend to discuss this phenomenon.
We thank the reviewer for this important comment. We have now expanded our discussion section to address this phenomenon. We specifically address the possibility of children having false-negative tests. We also address that children may other viral infections such as influenza, rhinovirus, respiratory syncytial virus or respiratory adenovirus- which may call for additional testing pathways and strategies. The revised discussion section can be viewed between lines 284-312.
- Spelling and grammar mistakes and table formats are addressed.
For more details please find the attached file with track changes

Reviewer 2 Report
adequate abstract. materials and methods well described, also results. References adequated. The study is interesting
Author Response
Thank you for your review and support, we appreciate your time to read the manuscript.